# Risk of Latent Tuberculosis Infection Reactivation in Patients Treated with Tumor Necrosis Factor Antagonists: A Five-Year Retrospective Study

**DOI:** 10.3390/tropicalmed10070190

**Published:** 2025-07-07

**Authors:** Işıl Deniz Alıravcı, Pınar Mutlu, Sibel Oymak, Ufuk Ilter Guney, Oguzhan Keskin

**Affiliations:** 1Department of Infectious Diseases and Clinical Microbiology, Çanakkale Onsekiz Mart University, Çanakkale 17020, Türkiye; dr_isildeniz@hotmail.com; 2Department of Chest Diseases, Çanakkale Onsekiz Mart University, Çanakkale 17020, Türkiye; ufukilter.guney@saglik.gov.tr (U.I.G.); oguzhan.keskin@saglik.gov.tr (O.K.); 3Department of Public Health, Çanakkale Onsekiz Mart University, Çanakkale 17020, Türkiye; cevizci.sibel@gmail.com

**Keywords:** TNF-α blockers, retrospective, latent tuberculosis infection, tuberculosis

## Abstract

Background: This study aims to reveal the demographic and clinical data of patients receiving TNF-α blockers, to compare the characteristics of those who received latent tuberculosis infection (LTBI) treatment and those who did not, and to evaluate and determine potential risk factors for developing active TB disease. Methods: A systematic retrospective study was conducted in a tertiary university hospital examining all patients receiving at least one TNF-α blocker between January 2019 and October 2024. The incidence of tuberculosis (TB) was analyzed across various TNF-α blocker medications in patients, both with and without LTBI treatment. Results: A total of 519 patients had TNF-α blockers: 452 (87.09%) underwent TST, 193 (37.1%) underwent booster TST, and 33 (6.3%) underwent IGRA/TST; 362 (69.7%) were treated for LTBI, and 7 (1.3%) developed TB. Comparing all TNF-α blockers, adalimumab showed a higher risk of TB. Patients with and without LTBI treatment did not significantly differ in TB incidence after biologic therapy. Conclusions: The incidence of TB in people taking TNF-α blockers was higher compared to the incidence in the general population. LTBI screening, including both TST and IGRA, should be performed with TST and IGRA tests, and LTBI-positive individuals should be started on preventive treatment. However, it should not be forgotten that active TB disease may also develop in LTBI-negative individuals.

## 1. Introduction

Tumor necrosis factor-alpha (TNF-α) is a homotrimeric transmembrane protein expressed by T lymphocytes, macrophages, fibroblasts, natural killer (NK) cells, and smooth muscle cells, which plays a role in inducing apoptotic cell death and inflammation, as well as inhibiting tumor development and viral replication. It plays a key role in cell proliferation, differentiation, and apoptosis during organogenesis [1]. TNF-α levels increase both locally and systemically in chronic inflammatory diseases such as psoriasis, rheumatoid arthritis (RA), ankylosing spondylitis (AS), and Crohn’s disease. It plays a key role in infections associated with granuloma formation by promoting the development and maintenance of granuloma structures [2].

Therefore, anti-TNF agents impair the granuloma-forming function of TNF-α, leading to increased susceptibility to granulomatous infections. The risk of infections such as Mycobacterium tuberculosis and Histoplasma capsulatum is increased due to this mechanism [3]. Additionally, anti-TNF therapy reduces macrophage activation and phagosome formation, thereby increasing susceptibility to intracellular pathogens and contributing to neutropenia-associated opportunistic infections. It has been shown that the risk of serious bacterial, fungal, viral, and parasitic infections is elevated in patients receiving anti-TNF therapy [4].

Currently, five anti-TNF agents are approved by the U.S. Food and Drug Administration (FDA) for use in inflammatory diseases: etanercept, infliximab, adalimumab, certolizumab, and golimumab. Infections such as tuberculosis (TB), hepatitis B virus (HBV), and hepatitis C virus (HCV) may occur as adverse effects of TNF-α inhibitor therapies [5].

Previous systematic reviews have reported an increased risk of TB in patients receiving TNF-α inhibitors [6], with notably higher rates observed in Asia and South America compared to Western Europe and North America [7]. Combined or sequential use of anti-TNF agents with other immunosuppressive drugs increases the risk of TB more than monotherapy [8]. Although the timing of TB onset varies by treatment modality, extrapulmonary TB is observed in more than half of cases, followed by pulmonary and disseminated forms [9]. Patients with a history of TB may experience reactivation under anti-TNF therapy, even after previous TB treatment [8].

The initiation of anti-TNF therapy is contraindicated in the presence of active TB [10]. Therefore, active TB must be excluded prior to treatment. In Turkey, patients diagnosed with TB are managed according to the National Tuberculosis Diagnosis and Treatment Guidelines. As stated in the national guideline, screening for latent TB infection (LTBI) is mandatory prior to the initiation of anti-TNF therapy, and patients with active TB must be excluded.

In Turkey, the tuberculin skin test (TST) or interferon-gamma release assay (IGRA) is used for initial and follow-up LTBI screening. If the TST result is ≥5 mm, it is considered positive, and active TB is ruled out before initiating a 9-month isoniazid (INH) prophylaxis for LTBI. In cases with <5 mm induration, a repeat (booster) TST or IGRA is performed within 1–3 weeks. If the IGRA is positive or the second TST is ≥5 mm, LTBI prophylaxis is initiated. Due to increased TST reactivity, IGRA is preferred in psoriasis patients who are candidates for anti-TNF therapy [11].

INH is used for 9 months as prophylaxis, while rifampin (RIF) for 4 months is recommended when INH cannot be used. Prophylaxis should be initiated at least 1 month before starting anti-TNF treatment [11]. If TB occurs during anti-TNF therapy or within 6 months after discontinuation, it is considered therapy-related [12].

In Turkey, a country with intermediate TB prevalence, screening and management of LTBI prior to initiating TNF-α antagonists is of critical public health importance. Understanding the demographic and clinical characteristics of these patients and identifying risk factors for TB development can help improve preventative strategies.

The aim of this cross-sectional retrospective study is to evaluate the incidence of TB in patients treated with TNF-α blockers, to compare outcomes between those who did and did not receive LTBI treatment, and to identify risk factors associated with TB reactivation.

## 2. Materials and Methods

Study Population and Data Source: A total of 519 adult patients who received at least one anti-TNF-α treatment and were referred from physiotherapy and rehabilitation, dermatology, ophthalmology, and gastroenterology clinics between January 2019 and October 2024 were retrospectively included in the study. Data were obtained by reviewing patient files through the electronic medical records system of Çanakkale 18 Mart University Hospital, a tertiary referral center and regional tuberculosis reference hospital. There were no restrictions regarding age, treatment duration, or length of follow-up. Underlying diagnoses included rheumatoid arthritis (RA), spondyloarthropathies, unclassified inflammatory rheumatic diseases, inflammatory bowel diseases (IBDs), uveitis, dermatologic disorders, and other chronic inflammatory conditions. Patients were categorized based on whether they received LTBI prophylaxis prior to the initiation of TNF-α blocker therapy.

Demographic and clinical characteristics of the groups (with and without LTBI treatment) were compared. In addition, patients who developed active TB under anti-TNF treatment were identified and analyzed in detail. Each patient may have received more than one TNF-α blocker, and many were concurrently on additional immunosuppressive agents.

The distribution of TNF-α blockers among the patients was as follows: adalimumab: 255 patients; infliximab: 82 patients; certolizumab: 77 patients; etanercept: 54 patients; golimumab: 51 patients.

Definition of LTBI and TB: The definition of latent tuberculosis infection (LTBI) and the screening procedures prior to anti-TNF treatment were based on the National Tuberculosis Diagnosis and Treatment Guideline of Türkiye. Patients with TST induration ≥ 5 mm were considered positive for LTBI and were administered prophylactic treatment. Patients with TST induration < 5 mm underwent either a booster TST within 1–3 weeks or IGRA (interferon-gamma release assay) testing. Prophylactic LTBI treatment was initiated in all patients with a positive IGRA or a booster TST result of ≥5 mm. Patients with negative IGRA or booster TST results, along with normal physical examination and chest X-ray, did not receive prophylactic LTBI treatment.

Patients receiving anti-TNF treatment were monitored every 6 months, regardless of symptoms, for signs of active TB, including anamnesis, physical examination, and radiological evaluation [11]. Clinical notes from these evaluations were extracted from the hospital system and analyzed.

Statistical Analysis: All statistical analyses were performed using SPSS (Statistical Package for the Social Sciences) software (IBM SPSS Statistics, version 26, IBM Corp., Armonk, NY, USA). Categorical variables were summarized as frequencies and percentages (%), while continuous variables were presented as mean ± standard deviation (SD) or median [minimum–maximum], as appropriate.

Chi-square or Fisher’s exact test was used for comparing categorical variables. Normality of continuous variables was assessed using the Shapiro–Wilk test. Comparisons between independent groups were made using the Student’s t-test for normally distributed variables.

To compare the risk of TB development between adalimumab users and other TNF-α blockers, the risk ratio (RR) was calculated. The RR was determined by comparing the incidence of TB in the adalimumab group versus the non-adalimumab anti-TNF group. A *p*-value of <0.05 was considered statistically significant.

## 3. Results

### 3.1. Baseline Characteristics of All Patients Received TNF-α Blockers

A total of 519 patients were included during the 5-year study period. Table 1 presents the baseline characteristics of the study population. The mean age was 48.7 ± 13.51 years. The majority of patients (52.7%) were diagnosed with ankylosing spondylitis.

Among the participants, 32.9% remained on their initial TNF-α inhibitor, whereas 28.5% switched at least once to another anti-TNF agent. Our data showed that adalimumab was the most frequently prescribed biologic agent, used by 49.1% of patients.

Approximately 27.2% of the patients had received methotrexate prior to starting biologic therapy. Among the 519 patients, 5 (1.38%) discontinued anti-TNF treatment due to interruption, 25 (4.8%) discontinued on their own, 13 (2.5%) discontinued based on the physician’s decision, and 43 (8.2%) were lost to follow-up. Additionally, 20 patients (3.8%) reported side effects related to anti-TNF therapy.

Of the 519 patients, 452 (87.1%) underwent TST, 193 (37.2%) underwent booster TST, 98 (18.8%) underwent IGRA, and 33 (6.3%) underwent both TST and IGRA testing. Among the 452 patients tested with TST, 250 (55.3%) had a positive result. Of the 193 patients who underwent booster TST, 64 (33.1%) were positive. Among 98 patients tested with IGRA, 41 (41.8%) had a positive result. Among 33 patients who received both TST and IGRA, 12 (36.3%) were positive (Table 1)

### 3.2. Comparison of Patients With and Without Treatment for LTBI

Table 2 presents a comparison between patients who received LTBI treatment and those who did not. The LTBI-positive group (*n* = 362) and LTBI-negative group (*n* = 157), with statistically significant differences observed (*p* < 0.001) of TNF-α inhibitor users differed significantly in terms of sex distribution. Female sex was significantly more common in the LTBI-negative group, whereas male sex was predominant in the LTBI-positive group.

The percentage of ankylosing spondylitis was higher among LTBI-positive patients compared to LTBI-negative patients. The mean duration of anti-TNF therapy was 4.05 ± 3.47 months in LTBI-positive patients and 3.47 ± 3.13 months in LTBI-negative patients.

A statistically significant association was observed between uveitis diagnosis and absence of LTBI treatment; uveitis was more common in patients who did not receive LTBI treatment. Although the prevalence of malignancy was higher among those receiving LTBI prophylaxis, the difference was not statistically significant.

The overall percentage of TNF-α inhibitor users was higher in the LTBI-positive group (69.7%) compared to the LTBI-negative group (30.2%). Although the proportion of patients treated with adalimumab was higher in the LTBI-negative group (59.8%) compared to the LTBI-positive group (44.4%), no statistically significant difference was detected. The number of patients receiving infliximab (*n* = 66, 18.2%) was greater in the LTBI-positive group, although the difference was not statistically significant.

### 3.3. LTBI Treatment Compliance of the Patients

Table 3 shows the LTBI treatment compliance of the patients. The mean duration of LTBI treatment with INH was 8.6 months. Of 362 patients who started LTBI treatment, 338 [93.3%] patients were able to complete LTBI treatment, and six [1.65%] of them exchanged because of adverse effects. Hepatotoxicity occurred in 12 [3.31%] patients. Hepatotoxicity was defined as either transaminitis (an increase in alanine aminotransferase and/or aspartate aminotransferase above the laboratory upper limit of normal) or hyperbilirubinaemia (an increase in total bilirubin level above 34.2 μmol/L, or a combination of both).

### 3.4. Patients Who Developed Active Tuberculosis

Table 4 shows the clinical characteristics of the seven patients who developed active TB disease during anti-TNF therapy. Four (57.1%) of them experienced pulmonary TB, and three (42.8%) experienced extrapulmonary TB. During the 5-year study duration, 5 of the 362 LTBI-positive patients and 2 of the 157 LTBI-negative patients developed active TB disease. The mean age of patients who developed active TB disease was 58.2 years, and the male gender was predominant. These patients were diagnosed with AS, psoriasis, and RA; no patient with a diagnosis of malignancy developed active TB disease. The median onset of active TB disease in patients was 4.1 months. Four patients had their anti-TNF treatment terminated; treatment information for two patients was not available, and death occurred in one patient.

Of the seven patients who developed active TB disease, four had received adalimumab, and the remaining three patients had received etanercept, infliximab, and certolizumab, respectively. No patient who received golimumab alone developed active TB disease. Of the seven patients, four were receiving adalimumab (adalimumab RR: 0.721, *p* = 0.720, 95% CI: 0.160–3.255) (Table 5) and five were undergoing additional immunosuppressive therapy (RR: 3.744, *p* = 0.125, 95% CI: 0.720–19.481) (Table 6). No statistically significant differences were found.

Of the seven patients who developed active TB disease, only two (28.5%) had positive TST results, one (14.2%) had a positive booster TST result, two (28.5%) had positive IGRA results, and none had positive TST + IGRA results before anti-TNF treatment.

## 4. Discussion

Our findings are based on retrospective data and statistical associations; therefore, causal relationships cannot be confirmed. For example, the higher LTBI detection rates in ankylosing spondylitis patients may reflect the immunological nature of the disease, or a greater likelihood of initiating TNF-α blockers rather than a true predisposition. Similarly, the use of infliximab or methotrexate might be driven by disease severity or type, rather than being the cause of differential TB risk.

Tumor necrosis factor-alpha (TNF-α) is a cytokine produced by macrophages, T cells, and natural killer (NK) cells, existing in both transmembrane and soluble forms, and plays a key role in regulating inflammation, cell survival, and apoptosis [13]. The use of anti-TNF therapies is increasing due to their demonstrated efficacy in managing chronic inflammatory diseases [14,15]. However, recent meta-analysis indicates that anti-TNF-α therapy does not significantly increase lymphoma risk in patients with rheumatoid arthritis (RR = 1.43; 95% CI: 0.59–3.47) [16].

Previous studies have shown that anti-TNF therapies increase the risk of active TB by approximately 1.6 to 25.1 times [17,18]. According to data from the Turkish Ministry of Health, the incidence of TB in 2020 was 10.6 per 100,000 population [19]. In our study, 7 of 519 patients (1.34%) receiving anti-TNF therapy developed active TB, indicating a 126-fold increase in TB incidence.

Other studies conducted in Turkey have reported active TB development rates between 0.85% and 1.5%, which is consistent with our findings [20,21,22,23]. In a 2023 study from Turkey, the rate of active TB among patients receiving anti-TNF therapy was 0.97%, similar to our result; however, that study used 2017 TB incidence data (14.6 per 100,000), yielding a 66-fold increased risk [23]. This highlights the importance of defining clear strategies to reduce active TB risk in TB-endemic countries.

In our study, two of the seven patients who developed active TB were LTBI-negative, representing 0.38% of the entire cohort and 1.27% of LTBI-negative patients. This aligns with prior research showing that active TB can occur even in patients with negative screening results [24,25,26]. Possible explanations include concurrent immunosuppressive therapies, reduced test sensitivity, BCG vaccine cross-reactivity, or new TB exposure in medium-to-high burden regions. Therefore, clinicians should maintain a high index of suspicion for active TB in all patients receiving anti-TNF therapy, regardless of LTBI screening outcomes.

In our study, hepatotoxicity developed in 12 patients (3.31%) receiving LTBI prophylaxis, but none required treatment discontinuation. This aligns with a recent meta-analysis, which reported an overall isoniazid-induced hepatotoxicity rate of approximately 2.6% in patients treated for LTBI [27]. In a study involving 289 patients who started INH for LTBI prophylaxis, 49 (17%) had elevated AST and/or ALT levels. The relatively high percentage was attributed to mild (<3-fold) enzyme elevations without clinical symptoms, and no patient required INH discontinuation [28]. In a meta-analysis including patients undergoing dialysis, organ or hematologic transplant, or with silicosis, the INH-related hepatotoxicity rate was 2.6%, considered clinically acceptable [29].

A meta-analysis involving 98,483 patients treated with at least one anti-TNF agent reported 947 cases of active TB, of which 62.2% were pulmonary TB [7]. In our study, four of seven patients (57.1%) who developed active TB had pulmonary involvement, which is consistent with the literature [30,31].

The risk of developing TB varies among TNF-α inhibitors. Some studies suggest higher TB risk with infliximab and adalimumab compared to etanercept, especially in regions with high TB prevalence [32], while others report no significant difference in TB risk between infliximab and etanercept in RA patients [33]. Data remain limited for certolizumab and golimumab. In our study, four of seven TB cases were among adalimumab users. Although this suggests a possible association, the difference was not statistically significant, likely due to the limited sample size.

The choice between IGRA and TST for LTBI screening depends on factors such as accuracy, cost, and population characteristics. While both have advantages, IGRA offers greater specificity and similar sensitivity, particularly in BCG-vaccinated populations such as Turkey [34,35]. TST remains more cost-effective in low-resource settings but suffers from reading variability and lower specificity [35]. IGRA, although costlier, reduces false positives and may be more cost-effective long-term by preventing unnecessary treatment [35].

In our study, IGRA alone was used in 98 patients (18.88%), and both TST and IGRA were used in 33 patients (6.35%). We suggest that IGRA may be more cost-effective for LTBI screening in patients undergoing anti-TNF therapy, especially in those with concomitant immunosuppression or skin conditions such as psoriasis.

## 5. Conclusions

Limitations: The retrospective design, lack of standardized diagnostic tools, and reliance on hospital-based records are study limitations. Sample size calculation and multivariate analysis were not performed, which limits the generalizability of the findings.

Conclusion: This study highlights that LTBI screening and monitoring remain essential for patients receiving TNF-α inhibitors, especially in TB-endemic regions. Clinicians should remain vigilant, even in LTBI-negative individuals. Future prospective multicenter studies using standardized diagnostic protocols are warranted.

## Figures and Tables

**Table 1 tropicalmed-10-00190-t001:** Baseline characteristics of the patients who started anti-TNF biologic therapy (*n* = 519).

Variables	*n*	%
Age, years		
15–44	208	40.07
45–64	239	46.05
65–86	72	13.88
Sex, female	272	52.40
Diagnosis (prevalence in the sample) †		
Ankylosing spondylitis (AS)	274	52.79
Rheumatoid arthritis (RA)	93	17.91
PSA	59	11.36
Psoriasis	41	7.89
Uveitis	30	5.78
Crohn’s/ulcerative colitis	26	5.00
Fibromyalgia syndrome (FMS)	3	0.57
Behçet	6	1.15
SLE	1	0.19
Malignancy	11	2.11
Mortality	2	0.38
Biologic agent		
Adalimumab	255	49.13
Etanercept	54	10.40
Golimumab	51	9.83
İnfliximab	82	15.80
Certolizumab	77	14.84
Biologic agent duration (SD) (years)	4.05	3.47
Previous anti-TNF treatments		
None	200	38.53
One treatment	171	32.94
Different treatments	148	28.51
Anti-TNF treatment outcome		
Interrupted treatment of anti-TNF	5	1.38
Discontinued by patient	25	4.81
Discontinued by physician	13	2.50
Loss to follow-up	43	8.28
Additional immunosuppressive treatment		
Methotrexate	141	27.17
Others	69	13.29
None	309	59.54
Adverse effects of anti-TNF treatment		
Not observed	499	96.15
Observed	20	3.85
Tests for latent tuberculosis		
Tuberculin skin test (TST)	452	87.09
Booster TST	193	37.18
Interferon gamma release assay (IGRA)	100	19.26
TST + IGRA	33	6.35
TST ^a^	250	55.30
TST ^b^	64	33.16
IGRA ^c^	41	41.83
TST + IGRA ^d^	12	36.36
Treatment for LTBI	362	69.75
Treatment for TBI	7	1.34

*n*: Number of participants. %: Frequency in percentage of column. †: Some participants had more than one diagnosis. The table shows the prevalence of the diagnoses in the participants. ^a^ Positive TST (≥5 mm of induration). ^b^ Positive booster TST (≥5 mm of induration). ^c^ Positive IGRA. ^d^ Positive TST + IGRA.

**Table 2 tropicalmed-10-00190-t002:** Comparison of patients with and without treatment for LTBI.

Variables (*n*, %)	Treatment for LTBI		*p*
	Received (*n* = 362)	Not Received (*n* = 157)	
Age, years	47.21	49.42	0.114 *
	*n* (%)	*n* (%)	
Sex			
Female	173 (47.79)	99 (63.05)	0.001 **
Male	189 (52.20)	58 (36.94)	
Diagnosis †			
Ankylosing spondylitis	218 (60.22)	56 (35.66)	<0.001 **
Rheumatoid arthritis	58 (16.02)	35 (22.29)	0.087 **
PSA	41 (11.32)	18 (11.46)	0.963 **
Psoriasis	25 (6.90)	16 (10.19)	0.203 **
Uveitis	11 (3.03)	19 (12.10)	<0.001 **
Crohn’s/ÜK	14 (3.86)	12 (7.64)	0.070 **
FMS	1 (0.27)	2 (1.27)	0.219 ***
Behçet	5 (1.38)	1 (0.63)	0.673 ***
SLE	0 (0.00)	1 (0.63)	0.303 ***
Malignancy	9 (2.5)	2 (1.3)	0.518 ***
Biologic agent	362 (69.7)	157 (30.2)	
Adalimumab	161 (44.47)	94 (59.87)	0.012 **
Etanercept	39 (10.77)	15 (9.55)	
Golimumab	41 (11.32)	10 (6.36)	
İnfliximab	66 (18.23)	16 (10.19)	
Certolizumab	55 (15.19)	22 (14.01)	
Mean duration of anti-TNF treatment (years ± SD)	4.42 ± 3.55	3.22 ± 3.13	<0.001 *
Additional immunosuppressive treatment			
Methotrexate	95 (26.24)	46 (29.29)	0.194 **
Others	43 (11.87)	26 (16.56)	
None	224 (61.87)	85 (54.14)	
Adverse effects of anti-TNF treatment			
Not observed	347 (95.85)	152 (96.81)	0.602
Observed	15 (4.14)	5 (3.18)	
Those who develop active tuberculosis (*n* = 7)	5 (1.38)	2 (1.27)	0.642
Pulmonary TBC (*n* = 4)	3 (60.00)	1 (50.00)	0.809
Extrapulmonary TBC tb (*n* = 3)	2 (40.00)	1 (50.00)	0.809

*n*: Number. %: Frequency in percentage of column. *: *t*-Test. **: Chi-square. ***: Fisher’s exact test. †: Some participants had more than one diagnosis. The table shows the prevalence of the diagnoses in the participants.

**Table 3 tropicalmed-10-00190-t003:** LTBI treatment compliance of the patients.

Variable	*n*	%
Started treatment for LTBI	362	100
Finished treatment for LTBI	338	93.37
Exchanged treatment for LTBI	6	1.65
LTB treatment duration mean value (SD) (months)	8.68	1.27
Death	2	0.55
Hepatotoxicity	12	3.31

*n*: Number. %: Frequency in percentage of column.

**Table 4 tropicalmed-10-00190-t004:** Patients who developed active tuberculosis.

Variables (*n*: 7)	*n*	%
Mean Age	58.2	
Sex		
Female	3	57.15
Male	4	42.85
Diagnosis		
AS	2	28.57
RA	2	28.57
Psoriasis	2	28.57
RA + Crohn	1	14.28
Biologic agent		
Adalimumab	4	57.14
Etanercept	1	14.29
Infliximab	1	14.29
Certolizumab	1	14.29
Biologic agent duration (years)	4.14	3.07
Additional immunosuppressive treatment		
MTX	4	57.14
Siklosporin	1	14.28
None	2	28.57
Positive TST number	2	28.57
Positive booster TST number	1	14.28
IGRA positive	2	28.57
TST + IGRA positive	0	0.00
Received LTBI treatment	5	71.42
Non-received LTBI treatment	2	28.57

**Table 5 tropicalmed-10-00190-t005:** Risk ratio of patients receiving adalimumab developing active tuberculosis.

	Patients with Active Tuberculosis	Healthy Group	*p*
Adalimumab	4 (57.14%)	251 (49.02%)	0.720
Others	3 (42.85%)	261 (50.97%)	

**Table 6 tropicalmed-10-00190-t006:** Risk ratio of patients receiving immunosuppression developing active tuberculosis.

	Patients with Active Tuberculosis	Healthy Group	*p*
Immunosuppression positive	5 (71.42%)	205 (40.03%)	0.125
Immunosuppression negative	2 (28.57%)	307 (59.96%)	

## Data Availability

The data presented in this study are available on request from the corresponding author. The data are not publicly available due to ethical restrictions and the confidentiality agreements with the institution regarding patient health information.

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
