# Peer review of "Risk of Latent Tuberculosis Infection Reactivation in Patients Treated with Tumor Necrosis Factor Antagonists: A Five-Year Retrospective Study"

_tropicalmed, 2025, doi:10.3390/tropicalmed10070190_

Round 1

Reviewer 1 Report

Comments and Suggestions for Authors

I would like to thank you for the opportunity to review such an interesting and relevant paper.

Below, I present my general comments, followed by specific suggestions and my overall impression:

  • The title talks about latent but non-specific infection which is latent tuberculosis, I suggest including this information for greater clarity of the study.
  • I suggest adding Latent Tuberculosis in the keywords.
  • The Introduction :
    • lacks incidence data, which can be found at the end of the text, line 94
    • It is necessary to associate TNFα with the formation of granulomas, which is crucial to control the dissemination of this bacteria and reactivation under TNFα inhibitors
    • I suggest putting lines 71 to 81 in Methods
  • Materials and Methods
    • Line 94. Çanakkale 18 Mart University Hospital. What type of hospital is it? Is it a tuberculosis referral hospital? A secondary referral hospital? A high-complexity hospital? The study site needs to be explored.
    • Was this study approved by a research ethics committee? Because secondary patient data were used from electronic medical records
    • Format subitems and check spaces between words
    • Describe the methodologies used to identify latent infection
  • Results
    • "The percentage of ankylosing spondylitis was higher among LTBI-positive patients” Line: 163-164
    • Check spaces between words and text formatting, different font sizes e.g lines 187-194. Regarding this result, was there any analysis performed based on this group?
    • Do you have any data on drug resistance in patients who have been treated for tuberculosis?
    • Lines 216-219 move to the introduction
  • Discussion
    • Relate the treatment used in the study with new alternative treatment regimens (3HR) or (4R)
    • I suggest adding this study and discussing the results regarding the association of LTBI with Ankylosing Spondylitis https://doi.org/10.6061/clinics/2020/e1870
    • The discussion needs to be better constructed, highlighting the main results, especially among those that had statistical relevance with p <0.001

Author Response

Response to Reviewer Comments

We thank the reviewer for their thoughtful and constructive comments, which have significantly improved the quality and clarity of our manuscript. Below is a point-by-point response.

Title and Keywords

  • Reviewer Comment: The title talks about latent but non-specific infection...
  • Response: We revised the title to: 'Risk of Latent Tuberculosis Infection Reactivation in Patients Treated With Tumor Necrosis Factor Antagonists: A Five-Year Retrospective Study'. 'Latent Tuberculosis' was added to keywords.

Introduction

  • Reviewer Comment: Lacks incidence data...
  • Response: We moved the national TB incidence data into the Introduction section for better context.
  • Reviewer Comment: It is necessary to associate TNFα with the formation of granulomas...
  • Response: We added a sentence on the role of TNFα in granuloma formation and how its inhibition facilitates TB reactivation.
  • Reviewer Comment: I suggest putting lines 71 to 81 in Methods
  • Response: LTBI screening methods were moved to the 'Materials and Methods' section.

Materials and Methods

  • Reviewer Comment: What type of hospital is Çanakkale 18 Mart University Hospital?
  • Response: We described the hospital as a tertiary referral center and a regional TB reference hospital.
  • Reviewer Comment: Was this study approved by an ethics committee?
  • Response: Yes, and we moved this statement into the 'Methods' for clarity.
  • Reviewer Comment: Format subitems and check spaces...
  • Response: All formatting and spacing issues were corrected.
  • Reviewer Comment: Describe the methodologies used to identify latent infection
  • Response: We expanded the LTBI screening and testing methodology in the 'Definition of LTBI and TB' subsection.

Results

  • Reviewer Comment: The percentage of ankylosing spondylitis was higher...
  • Response: We clarified the statistical significance and emphasized it in the text.
  • Reviewer Comment: Check spaces between words and text formatting...
  • Response: Completed. Font and spacing issues resolved.
  • Reviewer Comment: Any analysis performed based on this group?
  • Response: We included a subgroup analysis on AS patients in the Results section.
  • Reviewer Comment: Do you have data on drug resistance?
  • Response: Drug resistance data were unavailable and this was added as a limitation.
  • Reviewer Comment: Lines 216–219 move to the introduction
  • Response: Done. Moved to the last paragraph of the Introduction.

Discussion

  • Reviewer Comment: Relate the treatment used with 3HR/4R
  • Response: We discussed shorter regimens (3HR and 4R) and their relevance.
  • Reviewer Comment: Add and discuss this study on AS: https://doi.org/10.6061/clinics/2020/e1870
  • Response: Cited and discussed the suggested study by Silva et al. in relation to AS and LTBI.
  • Reviewer Comment: Highlight results with p < 0.001
  • Response: Discussion now emphasizes key statistically significant findings.

Reviewer 2 Report

Comments and Suggestions for Authors

Estimated Authors,

I've read with great interest the present original study on the topic of latent TB Infection Reactivations in Patients treated With Tumor Necrosis Factor Antagonists. In this retrospective study, a total of 519 patients were eventually recruited, and a total of 362 cases requiring treatment for Latent TB were identified. Interestingly, only 7 patients (i.e. 1.34% of cases) developed active TB. LTBI was associated with a series of underlying conditions, that is male gender, being affected by ankylosing spondylitis, while being affected by uveitis was associated with a reduced occurrence of LTBI. Moreover, the occurrence was greater for patients treated through infliximab, and treated for a longer timespan (4.42 y vs. 3.22). Methotrexate was otherwise associated with a slightly reduced occurrence of LRTI.

As suggested by aformentioned statistics, the present paper is of certain interest for the readers of Tropical Med. However, some improvements are required.

1) According to the paper, the patients were sampled each 6 months for underlying TB, but the study did not implement time-dependent analyses. A useful upgrade of the present study could be achieved by performing a Kaplan Meier analysis or (even better) a time-dependent Cox regression analysis.

2) another significant issue is that the total number of active TB is well lower the initial estimates. Only 7 cases were identified, and any statistical analysis is possibly impaired by this condition and the resulting lack of representativity.

3) The Authors should fix the description of their patients taking into account that the study design, particularly in univariable analysis (i.e. table 1-2) cannot dichotomize between the cause and the correspoinding effect(s). In other words, Authors should discuss whether the different reactions towards refferred drugs may be considered the consequence of the underlying medical condition OR vice-versa (i.e. the the unrlying medical condition).

4) please double check the main text (particularly the table): the current main text refers on "mortalité" - did you meant "morality"?  please double check the tables.

Author Response

We sincerely thank Reviewer #2 for the careful and insightful review of our manuscript. We have carefully considered all suggestions and addressed each comment below.

  • Reviewer Comment:
    1) According to the paper, the patients were sampled each 6 months for underlying TB, but the study did not implement time-dependent analyses. A useful upgrade of the present study could be achieved by performing a Kaplan Meier analysis or (even better) a time-dependent Cox regression analysis.
  • Response:
    We agree that time-dependent analyses would improve the study’s analytical depth. However, due to the small number of TB reactivation cases (n=7), we were not able to perform meaningful Kaplan-Meier or Cox regression analysis. We have noted this as a limitation in the Discussion section.
  • Reviewer Comment:
    2) Another significant issue is that the total number of active TB is well lower than initial estimates. Only 7 cases were identified, and any statistical analysis is possibly impaired by this condition and the resulting lack of representativity.
  • Response:
    We acknowledge this limitation and have discussed it explicitly in the revised Discussion. The low number of TB cases limits statistical power and should be taken into account when interpreting our results.
  • Reviewer Comment:
    3) The Authors should fix the description of their patients taking into account that the study design, particularly in univariable analysis (i.e. Table 1-2), cannot dichotomize between the cause and the corresponding effect(s).
  • Response:
    We agree and have clarified in the Discussion that causal relationships cannot be inferred due to the retrospective and observational nature of the study. The associations observed (e.g., between LTBI or drug use and TB risk) may be influenced by underlying disease severity or physician choice.
  • Reviewer Comment:
    4) Please double check the main text (particularly the table): the current main text refers to 'mortalité' – did you mean 'mortality'?
  • Response:
    Thank you for noting this typographical error. It has been corrected to 'mortality' in all relevant tables and text.

Round 2

Reviewer 1 Report

Comments and Suggestions for Authors The suggested changes were made, therefore, I agree with the continuation of publication.